# In Vitro Rearing Changes Social Task Performance and Physiology in Honeybees

**DOI:** 10.3390/insects13010004

**Published:** 2021-12-21

**Authors:** Felix Schilcher, Lioba Hilsmann, Lisa Rauscher, Laura Değirmenci, Markus Krischke, Beate Krischke, Markus Ankenbrand, Benjamin Rutschmann, Martin J. Mueller, Ingolf Steffan-Dewenter, Ricarda Scheiner

**Affiliations:** 1Biocentre, Department of Behavioural Physiology and Sociobiology, Julius-Maximilians-Universität Würzburg, Am Hubland, 97074 Würzburg, Germany; lioba.hilsmann@uni-wuerzburg.de (L.H.); lisa.rauscher@stud-mail.uni-wuerzburg.de (L.R.); laura.degirmenci@uni-wuerzburg.de (L.D.); ricarda.scheiner@uni-wuerzburg.de (R.S.); 2Department of Pharmaceutical Biology, Julius-von-Sachs-Institute, Julius-Maximilians-Universität Würzburg, Julius-von-Sachs-Platz 2, 97082 Würzburg, Germany; krischke@biozentrum.uni-wuerzburg.de (M.K.); martin.mueller@biozentrum.uni-wuerzburg.de (M.J.M.); 3Biocentre, Department of Animal Ecology and Tropical Biology, Julius-Maximilians-Universität Würzburg, Am Hubland, 97074 Würzburg, Germany; beate.krischke@uni-wuerzburg.de (B.K.); benjamin.rutschmann@uni-wuerzburg.de (B.R.); ingolf.steffan@uni-wuerzburg.de (I.S.-D.); 4Center for Computational and Theoretical Biology (CCTB), Julius-Maximilians-Universität Würzburg, Klara-Oppenheimer-Weg 32, 97074 Würzburg, Germany; markus.ankenbrand@uni-wuerzburg.de

**Keywords:** honeybee, artificial rearing, behavior, in vitro, juvenile hormone, triglycerides, PER, foraging, nursing

## Abstract

**Simple Summary:**

The rearing of honeybee larvae in the laboratory is an important tool for studying the effects of plant protection products or pathogens on developing and adult bees, yet how rearing under artificial conditions affects the later social behavior and physiology of the honeybees is mostly unknown. We, here, show that honeybees reared in the laboratory generally had a lower probability for performing nursing or foraging tasks compared to bees reared under natural conditions in bee colonies. Nursing behavior itself appeared normal in in vitro honeybees. In contrast, bees reared in the laboratory foraged for a shorter period in life and performed fewer trips compared to bees reared in colonies. In addition, in vitro honeybees did not display the typical increase in juvenile hormone titer, which goes hand-in-hand with the initiation of foraging in colony-reared bees.

**Abstract:**

In vitro rearing of honeybee larvae is an established method that enables exact control and monitoring of developmental factors and allows controlled application of pesticides or pathogens. However, only a few studies have investigated how the rearing method itself affects the behavior of the resulting adult honeybees. We raised honeybees in vitro according to a standardized protocol: marking the emerging honeybees individually and inserting them into established colonies. Subsequently, we investigated the behavioral performance of nurse bees and foragers and quantified the physiological factors underlying the social organization. Adult honeybees raised in vitro differed from naturally reared honeybees in their probability of performing social tasks. Further, in vitro-reared bees foraged for a shorter duration in their life and performed fewer foraging trips. Nursing behavior appeared to be unaffected by rearing condition. Weight was also unaffected by rearing condition. Interestingly, juvenile hormone titers, which normally increase strongly around the time when a honeybee becomes a forager, were significantly lower in three- and four-week-old in vitro bees. The effects of the rearing environment on individual sucrose responsiveness and lipid levels were rather minor. These data suggest that larval rearing conditions can affect the task performance and physiology of adult bees despite equal weight, pointing to an important role of the colony environment for these factors. Our observations of behavior and metabolic pathways offer important novel insight into how the rearing environment affects adult honeybees.

## 1. Introduction

Eusociality and division of labour are key factors in a functioning honeybee colony. The colony strongly depends on the proper execution and appropriate timing of various tasks by its members [1,2]. Reproduction is in the hands of the queen and drones, while the other tasks are conducted by sterile, female honeybee workers. Young honeybees perform in-hive tasks such as brood care [2]. At about 20 days of age, honeybees leave the colony to collect resources for the colony, e.g., nectar and pollen [1,2]. However, this age-dependent behavioral shift is very plastic. Removing foragers from a colony can induce nurse bees to start foraging precociously [3]. Another factor influencing the transition from nursing to foraging is nutritional restriction. When honeybees are deprived of lipids, the number of precocious foragers increases [4]. Environmental stress, which can induce an increase in juvenile hormone (JH III) or the neurohormone and transmitter octopamine [5], is generally linked to earlier foraging behavior in adult honeybees. It has been shown that immune-stressed honeybees have significantly lower expression levels of the juvenile hormone esterase and the egg yolk precursor vitellogenin, which is used to produce brood food [6]. Another study showed that nurse bees infested with the varroa mite are less attracted to brood pheromone compared to uninfested honeybees [7]. How these environmental stressors can influence the colony on a developmental level is poorly understood.

In vitro rearing of honeybees is a common method for studying diverse questions related to environmental risk assessment [8], development [9], pathogens [10,11], pesticides [12] and behavior [13]. While the method itself has frequently been improved by making minor changes to the rearing protocol [14,15,16,17,18], very few studies have evaluated the “quality” of honeybees obtained through this protocol and their behavior as adults.

There are, naturally, huge differences in the social environment of honeybees developing in the colony and those reared in vitro. While bees reared in the colony were reported to be contacted by nurse bees up to 2785 times [19], bees reared in the laboratory are deprived of social contact and lack the chemical interactions and the regular tending of nurse bees [14]. Instead, they are provided with artificial food in a constant amount calculated for the average honeybee larva per day [15]. One recent study shows that honeybees reared in vitro on a diet equivalent to normal feeding by nurse bees were not only smaller but had smaller lateral mushroom body calyces than their sisters reared in the colony [9], which is probably a result of social deprivation. A reduced mushroom body size has even been reported for fruit flies following social deprivation [20]. This indicates that even the standard feeding protocol during in vitro rearing might act as a nutritional stressor during larval development. Nutritional stress during the fifth larval instar can increase starvation resistance compared to honeybee larvae fed on a normal diet [21]. However, the same treatment leads to higher JH III hormone titers in 24-h-old honeybees and in seven-day-old honeybees, indicating a faster maturation due to stress. At the same time, larval starvation decreased responsiveness to sugar in seven-day-old honeybees, which normally corresponds to low JH III titers. Pollen deprivation during larval development delays the onset of foraging and decreases mean foraging span and the lifespan of adult honeybee workers [22]. Nutritional deprivation during larval development can also influence adult ovary development. Thus, reduced pollen intake during the larval stages was shown to reduce ovary development in adult honeybees [23].

An interesting study addressing flight behavior of in vitro-reared honeybees demonstrated a lower metabolic rate in in vitro-reared bees compared to honeybees reared in the colony. The former could not fly as fast under tethered conditions [24]. Studies on the performance of in vitro-reared honeybees in a natural environment, however, are rare. It has been shown that in vitro-reared honeybees visited queen cells less frequently compared to their colony-reared sisters [13].

Our study aims to evaluate the behavioral performance of in vitro-reared honeybees in comparison to honeybees reared in the colony in a field study. We investigated how larval rearing can influence nursing performance and foraging flights—two of the most important tasks performed in the hive. Suboptimal food provisioning by humans during in vitro rearing compared to optimal feeding by nurse bees in the hive, as well as the utilization of yeast as a pollen substitute in artificial rearing, might lead to developmental stress of the larvae. This stress may later lead to a reduced nursing activity in adult bees, possibly reducing colony fitness in the long run [7]. A reduced foraging performance might lead to malnourishment of the entire colony, strongly reducing winter survival. We further investigated whether in vitro rearing affects physiological traits that are important for task performance. In addition to testing the effects of artificial rearing on weight, we measured titers of JH III. This developmental hormone increases during the transition from nursing to foraging, with foragers normally having the highest titers of JH III [25]. Changes in JH III titers may point towards a different tempo in adult maturation [26]. We also measured abdominal triglyceride levels, which are related to the metabolic states of honeybees performing different tasks and considered to be negatively affected by JH III titers [27]. Thus, young honeybees typically have high triglyceride levels stored in the fat body while foragers are lean. JH III is also negatively correlated with the egg yolk precursor vitellogenin, an important factor in nursing behavior [28]. According to the double repressor hypothesis, JH and vitellogenin inhibit each other [28], and both have an opposite effect on sucrose responsiveness [29]. Sucrose responsiveness normally increases with age so that nurse bees have a lower sucrose responsiveness than foragers [30,31]. Our experiments not only allow us to link the rearing condition of larvae to later task performance, but aim to unravel the impact of larval rearing condition on important hormonal and metabolic aspects of adult maturation.

## 2. Materials and Methods

### 2.1. In Vitro Rearing

One group of honeybees could develop naturally in the colony (control), while another group of honeybees (treatment) was reared in the laboratory under the standard feeding diet as established previously [15] (160 µL diet; Table 1). We controlled for age by caging each queen onto an empty comb three days prior to the day of larval emergence. Cages were freely passable for workers. Upon emergence, the larvae were transported into the lab, individually grafted and placed into small plastic cups (Weisel cups, Heinrich Holtermann KG; Brockel, Germany). These cups, placed in a 48-well plate, contained 20 µL of diet “A”. Well plates and cups were pre-heated to 35 °C and kept on 35 °C warm thermal mats (ThermoLux, Witte + Sutor GmbH, Murrhardt, Germany) while grafting. Afterwards, the well plates were transferred to an incubator maintained at 35 °C and 95% relative humidity (RH). Larvae were fed over six consecutive days according to Table 1. Once larvae transformed into pupae, they were transferred into fresh 48-well plates, transferred into a new incubator at 35 °C and 75% RH and left untouched, apart from sparse mortality check-ups, until emergence. For the group of bees reared in the colony, we removed frames with emerging brood and placed them in an incubator maintained at 35 °C and 75% humidity.

### 2.2. Comparison of Nursing Behavior

After emergence, in vitro-reared honeybees and colony-reared, newly emerged honeybees (“colony-reared controls”) were marked using a colored number plate (Opalith Classic Garnitur; Heinrich Holtermann KG; Brockel, Germany) and superglue (UHU^®^ Sekundenkleber blitzschnell Pipette; UHU GmbH & Co. KG; Bühl, Germany). After tagging, honeybees were transferred into cages (internal dimensions: 8 cm × 5 cm × 5 cm; three impenetrable walls and one wire-framed wall) containing 50% sugar water, normal tab water and a pollen source. These cages were left in an incubator overnight at 35 °C and 50% RH for the superglue to fully dry. Afterwards, the honeybees were integrated directly into a four-framed observation hive. Observations started the next day by removing the walls on the side of the observation hive and observing the honeybees through see-through Plexiglas walls.

Observations were conducted for four consecutive weeks. They started at 10:30 a.m. and finished at 2:30 p.m. every day. All four frames were scanned systematically in a pseudo-randomized order. Every visible honeybee bearing a number plate and putting its head into a brood cell was recorded. Every week, the brood area on the frames was marked to ensure that only nurse bees were recorded during the observations.

### 2.3. Comparison of Foraging Behavior

Newly emerged, in vitro-reared honeybees and colony-reared control honeybees were color-marked on the abdomen and tagged with a unique RFID (radio-frequency identification) chip (mic3-TAG 64-bit read only, carrier frequency: 13.56 MHz, Microsensys GmbH, Erfurt, Germany) using superglue. After tagging, honeybees were transferred into cages (same as above) containing 50% sugar water, normal tab water and a pollen source. These cages were left in an incubator overnight at 35 °C and 70% RH for the superglue to fully dry.

After the drying period, honeybees of all treatment groups were placed within the cages into a six-frame queen right mini-plus colony outfitted with two specifically designed scanners (MAJA Bundle Bee Identification System: iID 2000 ISO 15693 optimized, Microsensys GmbH). These scanners were positioned in front of the hive entrance so honeybees leaving or returning to the colony had to pass both scanners in a defined order. Data were acquired as previously established [32]. Only technical outliers were removed from the data sets. Data were excluded when the time interval between the first and second scans was larger than 300 s. Thus, we excluded foraging events when one of the scanners did not work. Additionally, only complete foraging trips were analyzed, i.e., when both events (leaving and returning to the hive) fulfilled our criteria. After one day, the cages were opened, and the tagged honeybees were able to freely move around the colony. This adaptation period was used to increase acceptance of the young honeybees once they had been released into the colony. Afterwards, the recordings began.

### 2.4. Weight, Juvenile Hormone, Triglycerides and Sucrose Responsiveness

Honeybees were treated in the same way as in Section 2.2. However, they were placed in cages into a single six-frame mini-plus colony as in Section 2.3. Every week, 15 honeybees of both groups were removed from the colony to perform various experiments.

Honeybees were immobilized on ice, mounted in metal holders and fed until saturation. After one hour of adjustment to the holders, the proboscis extension response (PER) experiment was conducted. The antennae of the honeybees were first touched with a droplet of water and, afterwards, sequentially with increasing sugar concentrations, as established previously, to determine the individual responsiveness to sucrose, with an intertrial interval of two minutes [31,33]. The occurrence of the proboscis extension was recorded for each stimulation of the antennae with water and the following sucrose concentrations with equal logarithmic distances: 0.1% sucrose, 0.3% sucrose, 1% sucrose, 3% sucrose, 10% sucrose and 30% sucrose. Afterwards, bees were individually weighed. Honeybees were then immobilized on ice for a second time and fixed with needles onto a Styrofoam plate. A 5 µL amount of hemolymph was extracted by piercing the cuticle in between the fourth and fifth abdominal segments using glass micro capillaries (servoprax^®^, A1 0115; servoprax GmbH, Wesel, Germany). The hemolymph was sampled for analyzing JH III titers. The 5 µL of hemolymph was blown out onto a parafilm surface using a thin hose and transferred into an Eppendorf Tube^®^ (1.5 mL) using a pipette. The tubes were then stored in a container filled with liquid nitrogen. JH III in hemolymph was analyzed by LC-MS/MS using a Waters Acuity ultra-high performance liquid chromatography system coupled to a Waters Micromass Quattro Premier triple quadrupole mass spectrometer (Milford, MA, USA), as described before [34].

To analyze the triglycerides (TGs) of the fat bodies, one half of a honeybee’s frozen fat body was crushed in a cooled mixer mill (MM 400, RETSCH GmbH, Haan, Germany) using zirconia beads. After a short centrifugation (Centrifuge 5424; Eppendorf, Hamburg, Germany), the triglycerides were extracted twice using chloroform (1 mL), methanol (0.5 mL) and two TAG standards (2.5 µg each, 10:0 TAG and 17:0 TAG). After mixing and centrifugation, the supernatant was collected and 0.88% aqueous KCl (0.75 mL) was added. After phase separation, the upper phase was discarded, and 0.25 mL methanol and 0.25 mL H_2_O were added to the lower phase containing the lipid extract. Afterwards, the lower phase was dried under reduced pressure using a rotational vacuum concentrator (RVC 2-25 CDplus, Martin Christ Gefriertrocknungsanlagen GmbH; Osterode am Harz, Germany) at 50 °C. The dried residue was dissolved in 100 µL isopropanol and frozen at −20 °C until analysis with a UPLC–qTOF-MS (Waters Corporation; Milford, MA, USA), as described before [35]. The data were analyzed using MassLynx™ software from Waters^®^. Only the ten most frequently appearing TGs were used for statistical analysis.

### 2.5. Data Analysis

Statistical analyses and graph construction were conducted using GraphPad Prism (GraphPad Software Inc., V8, San Diego, CA, USA) and R (4.1.2) and the R packages “glmm TMB” v. 1.1.2.3 [36], “lme4” v. 1.1–27.1 [37], “DHARMa” v. 0.4.4 [38] and “lsmeans” v. 2.30-0 [39]. Proportional data were analyzed with a Chi-square test using GraphPad Prism. A Shapiro–Wilk test was used to test the nursing data for normal distribution. Since data were not distributed normally, a Mann–Whitney U test was used to analyze the nursing data using GraphPad Prism. Data for the foraging performance were not distributed normally (Shapiro–Wilk test). Effects of rearing condition on foraging performance was investigated with a generalized linear mixed model (GLMM) with treatment as fixed factor and the four different colonies as random factor and nbinom2 family. Physiological data were not distributed normally. Effects of rearing condition on physiology was investigated with a GLMM with treatment and weeks as fixed factors and nbinom2 family. Post hoc analysis was conducted using Tukey multiple comparison tests.

## 3. Results

### 3.1. Comparison of Nursing Behavior

In vitro rearing strongly affected the proportion of the bees performing nursing tasks (Figure 1A). Colony-reared honeybees had a significantly higher proportion of bees performing nursing behavior compared to in vitro-reared honeybees (Figure 1A and Table 2). However, the individuals actually performing nursing tasks did not differ in their task performance between rearing conditions, i.e., onset of nursing behavior (Figure 1B and Table 2), termination of nursing behavior (Figure 1C and Table 2) and nursing span (Figure 1D and Table 2).

### 3.2. Foraging Behavior

In vitro rearing had a significant influence on the proportion of foragers (Figure 2A). A significantly higher proportion of colony-reared honeybees became foragers compared to in vitro-reared honeybees (Figure 2A and Table 2). We next focused on those bees of both groups which actually performed foraging flights.

Foraging onset was significantly affected by rearing condition (Figure 2B and Table 2). In vitro-reared bees started foraging significantly earlier compared to colony-reared bees. The end of foraging was similarly affected by rearing environment (Figure 2C and Table 2) with in vitro-reared bees finishing foraging earlier than colony-reared bees. Since the onset and termination of foraging were earlier in the in vitro-reared group (Figure 2B,C), it is not surprising that foraging span was also significantly shorter in this group (Figure 2D and Table 2). While the rearing environment did not affect the duration of the foraging trips (Figure 2E and Table 2), it significantly affected the number of trips a forager performed per day (Figure 2F and Table 2).

### 3.3. Honeybee Morphology, Physiology and Sucrose Responsiveness

Honeybee weight decreased significantly with age across groups (Figure 3A and Table 3). Rearing condition did not affect honeybee weight (Figure 3A and Table 3). Titers of JH III increased significantly in both rearing groups during the first four weeks of adult life (Figure 3B and Table 3). However, treatment had a significant effect on JH III titers (Figure 3B and Table 3). In vitro-reared bees displayed a significantly lower JH III titer than colony-reared honeybees in week 3 and week 4 (Table 3), i.e., around the time when colony bees transition to foraging. Total lipids also changed significantly with the age of the bees after placement in the colony (Figure 3C and Table 3). In both rearing groups, lipids decreased over the four experimental weeks. Rearing condition affected lipid levels (Figure 3C and Table 3). In vitro-reared honeybees had significantly lower lipid levels in week 4 (Table 3). Individual sucrose responsiveness, measured as GRS, increased significantly with age in both rearing groups (Figure 3D and Table 3) and was unaffected by rearing environment (Figure 3D and Table 3).

## 4. Discussion

This study investigated the influence of rearing environment on adult honeybee workers. Larvae were either raised in vitro in an established laboratory assay or in the colony under natural conditions. One main effect we observed concerned the likelihood that a worker bee performed nursing or foraging tasks. The likelihood for performing these tasks was significantly reduced in bees reared in vitro compared to naturally reared honeybees. Why fewer in vitro reared honeybees performed the tasks is unclear. One possible explanation is that naturally reared honeybees are more vital than in vitro-reared bees due to artificial rearing acting as a nutritional stressor. It has been shown that nutritional deficits during in vitro rearing can lead to under-developed honeybee workers [21], which, in turn, might lead to the adult honeybees not nursing or foraging. In general, there are two main differences between the two rearing conditions. First, naturally reared larvae are fed by nurse bees according to their direct needs which are communicated by a brood pheromone signal [40]. Feeding larvae in the laboratory is based on an estimate of how much food a larva needs on average during development. Individual larvae might differ in their need for food, resulting in a larger span of possibly over-fed or under-fed larvae. The resulting honeybee workers might be less vital compared to their colony-reared sisters, leading to worker bees that are less capable to perform specific tasks. Another explanation might be the absence of fresh pollen in the diet of in vitro-reared larvae. During the in vitro rearing protocol, larvae are fed with a food mixture combining royal jelly with various concentrations of sugars and yeast [14,15,41]. However, while yeast is normally used as a pollen substitute, real pollen is not used during in vitro rearing to avoid moulding. This could negatively influence honeybee vitality because pollen quality and diversity are important for honeybees health and survival [42,43]. Additionally, it has been shown that bumblebees and probably also honeybees depend on fatty acids from pollen sources [44]. Increasing or decreasing fatty acid concentrations due to a yeast substitute could detrimentally affect honeybee health or behavior.

However, nurse bees which actually performed nursing tasks did not differ between rearing conditions. Honeybees reared in vitro started and terminated nursing tasks at a similar age as their naturally reared sisters. Another study investigating nursing behavior of in vitro-reared honeybees showed that these visited queen cells less frequently compared to colony-reared honeybees [13]. Uncovering the effects of in vitro rearing on the quality of worker and queen brood care could be an interesting question for future studies.

Intriguingly, in vitro rearing significantly influenced foraging behavior. In vitro-reared honeybees started foraging significantly earlier than bees reared in the colony but stopped foraging earlier than colony-reared bees, thus experiencing a significantly shorter foraging duration in total. Additionally, in vitro-reared honeybees also flew significantly fewer trips per day compared to naturally reared honeybees, indicating that in vitro-reared honeybees are not as strong as naturally reared honeybees and may not work as effectively as the latter. The duration of foraging trips was not significantly affected by in vitro rearing. However, Figure 2E suggests that the majority of in vitro-reared honeybees seem to have rather shorter foraging trips, lasting around 10 min, while the majority of hive-reared bees had flight durations of around 25 min, such as has been reported in other studies on untreated honeybees [45,46,47]. This question certainly deserves a more detailed investigation, since individual variation in flight duration can be large, as is also suggested by Figure 2E. Another study investigating the flight performance of in vitro-reared honeybees [24] did not find any differences in the distance flown between the two rearing conditions, which correlates with our study not finding any differences in the foraging duration. However, they showed differences in the maximum speed during the second flight with colony-reared honeybees reaching higher velocities than in vitro-reared honeybees. These results also indicate that naturally reared honeybees seem to be stronger and more vital compared to in vitro-reared honeybees.

To further investigate whether the difference in vitality also results in a different morphology or physiology, we looked at weight, JH III, TGs and GRS. We did not find any differences in weight between in vitro-reared and naturally reared honeybees. These results support earlier studies [48]. However, other studies also found in vitro-reared honeybees to be smaller [9,49]. These different results might be an effect of individually different metabolic needs, as discussed above. JH III increased with age, as frequently observed in honeybees [50,51,52]. However, during the third and fourth weeks of adult maturation, in vitro-reared honeybees had significantly lower JH III titers compared to colony-reared honeybees. These results seem contradictory, especially because in vitro-reared honeybees started foraging earlier. However, in vitro-reared honeybees stopped foraging around 14 days of age on average. Therefore, it is possible that in vitro-reared honeybees simply stopped foraging and returned to the colony to perform other tasks. This would correlate with their lower JH III titers. It has been shown that reverted nurse bees (foragers that go back to nursing) also show lower JH III levels compared to same-aged foragers [53]. However, it could also mean that the sampled honeybees in week 4 never foraged and, therefore, had low JH III titers. Future studies should analyze both factors simultaneously to uncover why in vitro-reared honeybees show lower JH III titers.

Triglycerides generally increased until week two and decreased from week two until week four. This pattern is likely linked to the transition from nursing to foraging [54]. However, we also found a significant treatment effect in week four, with significantly lower TG levels of in vitro-reared honeybees compared to colony-reared honeybees. These results appear contradictory. Usually, foragers have high JH III titers and low TG levels and nurse bees show the opposite pattern [53,54]. However, in vitro-reared honeybees had a significantly lower probability of performing social tasks, so that the in vitro-reared bees whose hemolymph was sampled in week four may never have performed nursing and foraging tasks, thus displaying rather unusual TG titers, a conclusion further emphasized by the general activity during our observation experiments (Figure A1 in Appendix A). In addition, there seems to be no direct link between JH III titers and TG titers in in vitro-reared bees in week four.

Rearing conditions did not significantly influence GRS. However, a tendency can be observed that in vitro honeybees show a lower GRS than colony-reared honeybees (factor treatment *p* = 0.055, interaction treatment*weeks *p* = 0.13; Table 3). Similar effects have been found earlier. Three-week-old in vitro-reared honeybees were significantly less responsive towards sucrose than naturally reared honeybees [13]. These findings further support the hypothesis that our in vitro-reared foragers either reversed back to nursing behavior, or the honeybees analyzed during week three and four never started to forage because, in general, it was shown that nurse bees are less responsive to sucrose than foragers [31]. Future studies should investigate further the “fate” of in vitro-raised honeybees in the hive. Are they generally less likely to perform any task and rather serve as a “reserve” in the hive, or do they simply perform tasks which were not in the focus of our investigation?

## 5. Conclusions

Overall, we found that the probability for performing social tasks was significantly reduced by in vitro rearing, likely because these bees were slightly weaker, although their weight did not differ from colony-reared bees. The typical increase in JH III titers observed during the transition from hive tasks to foraging was much less pronounced in lab-reared bees, correlating with their lower likelihood of becoming a forager, indicating hidden physiological modifications due to in vitro rearing. Importantly, honeybees reared in vitro were still able to perform all tasks. Their foraging performance was slightly reduced compared to that of colony-reared bees and there was no difference in the performance of nursing tasks between both treatment groups. Our data thus show that the method of in vitro rearing is nonetheless suitable for investigating honeybee behavior and physiology, provided that comparisons between treatment groups are all based on in vitro-reared worker bees.

## Figures and Tables

**Figure 1 insects-13-00004-f001:**
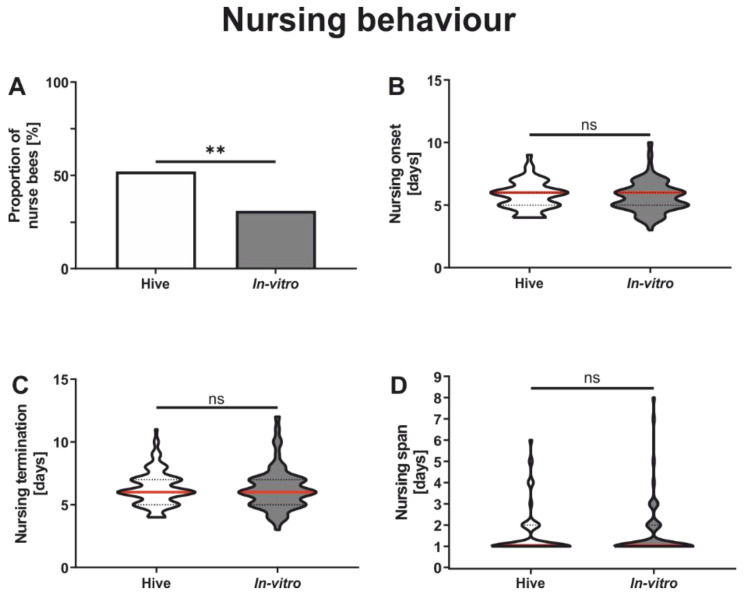
Influence of in vitro rearing on nursing behavior. (**A**): In vitro rearing significantly influenced the proportion of nurse bees. Significantly fewer in vitro-reared honeybees became nurses compared to colony-reared honeybees. (**B**): In vitro rearing did not affect the onset of nursing behavior. (**C**): In vitro rearing did not influence the termination of nursing behavior. (**D**): In vitro rearing did not affect nursing span. Significant differences between groups are indicated by asterisks (ns: *p* > 0.05, **: *p* < 0.01). For test statistics and sample size, see Table 2. Data in (**B**–**D**) display medians (red line) and 25% and 75% quartiles (lower and upper dotted lines, respectively).

**Figure 2 insects-13-00004-f002:**
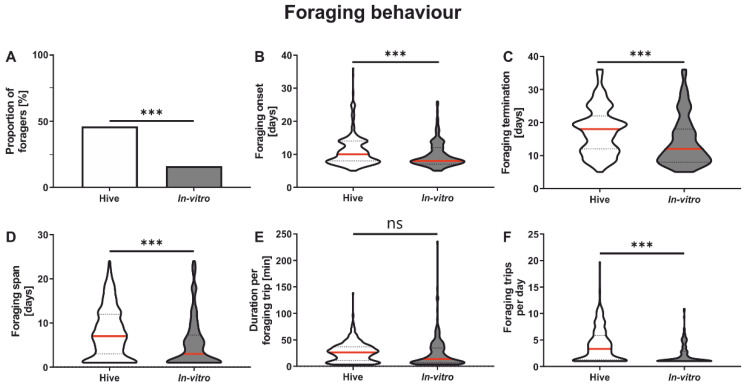
Influence of in vitro rearing on foraging behavior. (**A**): In vitro rearing significantly influenced the proportion of foragers. Significantly fewer in vitro-reared honeybees became foragers compared to colony-reared honeybees. (**B**): In vitro rearing significantly decreased the onset of foraging. (**C**): In vitro-reared honeybees terminated their foraging trips significantly earlier then colony-reared honeybees. (**D**): In vitro rearing significantly decreased the foraging span. (**E**): The duration per foraging trip was not influenced by the rearing environment. (**F**): In vitro-reared honeybees flew significantly fewer foraging trips per day then colony-reared honeybees. Significant differences between groups are indicated by asterisks (ns: *p* > 0.05, ***: *p* < 0.001). For test statistics and sample size, see Table 2. Data in (**B**–**F**) display medians (red line) and 25% and 75% quartiles (lower and upper dotted lines, respectively).

**Figure 3 insects-13-00004-f003:**
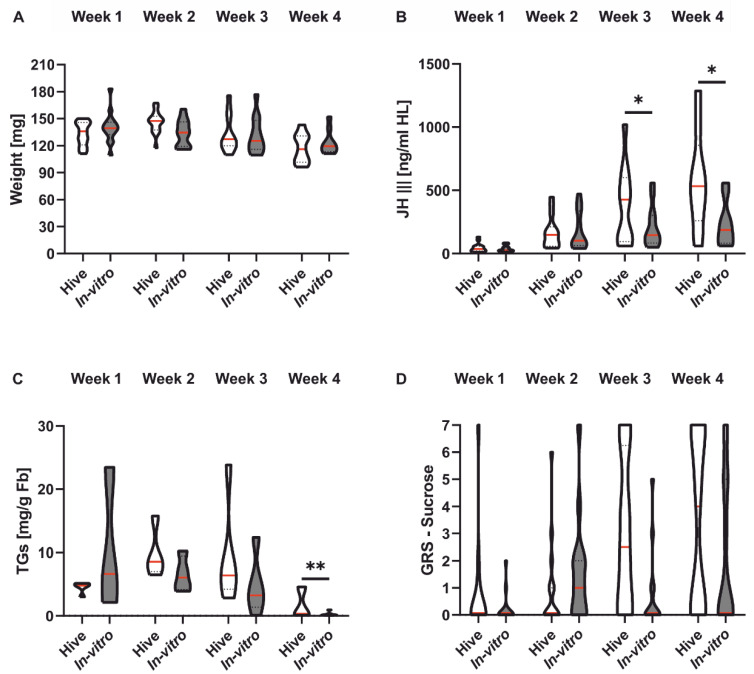
Body weight, JH III titers, TG levels and sucrose responsiveness of in vitro- and colony-reared honeybees in their first four weeks of life. (**A**): Honeybee body weight significantly decreased during the first four weeks of adult life. In vitro rearing did not influence honeybee weight. (**B**): Honeybee age significantly affected juvenile hormone tires (JH III), with older bees displaying higher JHIII titers. In addition, colony-reared bees had significantly higher JH III titers than in vitro-reared bees in week 3 and week 4. (**C**): Fat body triglyceride (TG) levels were significantly influenced by honeybee age, indicating a decrease in TG levels with increasing honeybee age. In vitro rearing significantly influenced TG titers in week 4. (**D**): Honeybee age significantly influenced the gustatory response scores (GRS). GRS generally increased with age. Rearing environment had no effect on GRS. For statistics, see Table 3. Significant differences between the two rearing groups are indicated by asterisks (*: *p* < 0.05; **: *p* < 0.01). The red line displays the median, and 25% and 75% quartiles are displayed by the lower and upper dotted lines, respectively.

**Table 1 insects-13-00004-t001:** Standard larval diet according to Aupinel et al. (2005). Feeding takes place over six days. On day one, larvae receive 20 µL of diet “A”. On day three, larvae receive 30 µL of diet “B”. On days four to six, larvae receive 30 µL, 40 µL, 50 µL, respectively.

Diet A	Royal Jelly	Fructose	Glucose	Yeast	Water
[%]	50	6	6	1	37
[g]	20	2.4	2.4	0.4	14.8
Diet B	Royal Jelly	Fructose	Glucose	Yeast	Water
[%]	50	7.5	7.5	1.5	33.5
[g]	20	3	3	0.6	13.4
Diet C	Royal Jelly	Fructose	Glucose	Yeast	Water
[%]	50	9	9	2	30
[g]	20	3.6	3.6	0.8	12

**Table 2 insects-13-00004-t002:** Test statistics for the analysis made in Figure 1 and Figure 2.

Analysis	Figure	Treatment	Sample Size	Test	Test-Value	*p*
Nursing proportion	1A	Colony	199	Chi-square	χ = 8.16	<0.01
		In vitro	192			
Onset of nursing	1B	Colony	104	Mann-Whitney	U = 3055	0.56
		In vitro	62			
Termination of nursing	1C	Colony	104	Mann-Whitney	U = 3098	0.67
		In vitro	62			
Nursing span	1D	Colony	104	Mann-Whitney	U = 3223	0.99
		In vitro	62			
Foraging proportion	2A	Colony	1016	Chi-square	χ = 21.04	<0.001
		In vitro	1005			
Onset of foraging	2B	Colony	472	GLMM	χ = 15.58	<0.001
Factor treatment		In vitro	164			
Termination of foraging	2C	Colony	472	GLMM	χ = 34.14	<0.001
Factor treatment		In vitro	164			
Foraging span	2D	Colony	472	GLMM	χ = 18.99	<0.001
Factor treatment		In vitro	164			
Duration per foraging trip	2E	Colony	472	GLMM	χ = 0.03	0.8834
Factor treatment		In vitro	164			
Foraging trips per day	2F	Colony	472	GLMM	χ = 54.31	<0.001
Factor treatment		In vitro	164			

**Table 3 insects-13-00004-t003:** Test statistics for the analysis made in Figure 3.

Analysis	Figure		Treatment	Sample Size	Test	Test Value	*p*
Weight	3A		Colony	63	GLMM	F = 0.06	0.804
Factor Treatment			In vitro	66			
Weight	3A		Colony	63	GLMM	F = 43.27	<0.001
Factor Week			In vitro	66			
Weight	3A		Colony	63	GLMM	F = 6.86	0.076
Interaction Treatment and Week			In vitro	66			
JH III	3B		Colony	50	GLMM	χ = 5.36	<0.05
Factor Treatment			In vitro	50			
JH III	3B		Colony	50	GLMM	χ = 146.70	<0.001
Factor Week			In vitro	50			
JH III	3B		Colony	50	GLMM	χ = 6.04	0.11
Interaction Treatment and Week			In vitro	50			
JH III	3B	Week 1	Colony	15	Tukey	*t* = 0.76	0.45
Pairwise Tukey Test			In vitro	15			
	3B	Week 2	Colony	15	Tukey	*t* = −0.30	0.77
			In vitro	14			
	3B	Week 3	Colony	11	Tukey	*t* = 1.99	<0.05
			In vitro	12			
	3B	Week 4	Colony	9	Tukey	*t* = 2.60	<0.05
			In vitro	9			
TGs	3C		Colony	17	GLMM	χ = 0.04	<0.05
Factor Treatment			In vitro	21			
TGs	3C		Colony	17	GLMM	χ = 81.12	<0.001
Factor Week			In vitro	21			
TGs	3C		Colony	17	GLMM	χ = 10.92	<0.05
Interaction Treatment and Week			In vitro	21			
TGs	3C	Week 1	Colony	5	Tukey	*t* = −1.43	0.16
Pairwise Tukey Test			In vitro	3			
	3C	Week 2	Colony	4	Tukey	t = 0.70	0.49
			In vitro	4			
	3C	Week 3	Colony	5	Tukey	*t* = 1.28	0.21
			In vitro	5			
	3C	Week 4	Colony	3	Tukey	*t* = 3.31	<0.01
			In vitro	9			
GRS	3D		Colony	17	GLMM	χ = 3.69	0.055
Factor Treatment			In vitro	21			
GRS	3D		Colony	17	GLMM	χ = 18.43	<0.001
Factor Week			In vitro	21			
GRS	3D		Colony	17	GLMM	χ = 5.68	0.13
Interaction Treatment and Week			In vitro	21			

## Data Availability

Correspondence and requests for materials should be addressed to F.S. and R.S.

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
