# Peer review of "In Vitro Rearing Changes Social Task Performance and Physiology in Honeybees"

_insects, 2021, doi:10.3390/insects13010004_

Round 1
Reviewer 1 Report
On page 3 lines 102-105 are about colony malnourishment caused by reduced nursing activity. Although the information about it is correct, such a situation can hardly be caused by in vitro raised bees (which you are experimenting on). Therefore this information is redundant and serves no purpose in text.
Some problems are caused by extreme data. Such thing can be seen in chart D with nursing span, chart F with foraging trips per day and also in chart E with duration per foraging trip. We know from our experience that outlying values are caused by bees, which for various reasons cannot return to the hive in time. (A bee flying for 250 minutes straight is hard to believe case.) Such values should be removed. (Especially extreme values about duration per foraging trip.)
From the shard E you can than clearly see that flights performed by in vitro bees from 25 to 50 minutest are in contrast with normal bees flights practically non-existent. This interesting phenomenon should be further explained in discussion, but it’s not.
Citations do not correspond with "Citations Style Guide". Using citations without a citation number cause chaos. Reader cannot easily find a citation by name, when the citation list is not in alphabetical order. Examples: lines 63,76,77,82,97,338.
Author Response
- On page 3 lines 102-105 are about colony malnourishment caused by reduced nursing activity. Although the information about it is correct, such a situation can hardly be caused by in vitro raised bees (which you are experimenting on). Therefore this information is redundant and serves no purpose in text.
We thank Reviewer 1 for the constructive feedback. We further explained the statement in lines 100 -104.
- Some problems are caused by extreme data. Such thing can be seen in chart D with nursing span, chart F with foraging trips per day and also in chart E with duration per foraging trip. We know from our experience that outlying values are caused by bees, which for various reasons cannot return to the hive in time. (A bee flying for 250 minutes straight is hard to believe case.) Such values should be removed. (Especially extreme values about duration per foraging trip.)
Thank you for bringing this valid point to our attention. We added an explanation on how we treated outliers in lines 173 – 177. However, we only removed technical outliers. We purposely included outliers that are most likely due to natural variation. While 250 min seems long for a foraging trip, it is not unheard of [1]. Removing outliers can also increase type I errors [2]. While we cannot be completely sure whether the remaining outliers are technical or natural outliers due to our sampling method, they were not detected by our strict criteria. Therefore, we left them in the data sets. However, these are very few data points which have no influence on the statistical analysis. In fact, only one bee in the in vitro group is responsible for the long peak in that group.
- From the shard E you can than clearly see that flights performed by in vitro bees from 25 to 50 minutest are in contrast with normal bees flights practically non-existent. This interesting phenomenon should be further explained in discussion, but it’s not.
Thank you very much for your feedback. We have now integrated a more detailed discussion of this interesting point in the discussion part (Lines 340 – 345). Our data show similar results for both groups and support findings of other studies observing foraging durations of honeybees [1,3,4]. Additionally, there is no statistical difference or even a tendency that both groups differ (Factor trip duration: p = 0.88).
- Citations do not correspond with "Citations Style Guide". Using citations without a citation number cause chaos. Reader cannot easily find a citation by name, when the citation list is not in alphabetical order. Examples: lines 63,76,77,82,97,338.
Thank you for bringing this to our attention. We removed these citations, wherever possible.
- Cho, Y.; Jeong, S.; Lee, D.; Kim, S.; Park, R.J.; Gibson, L.; Zheng, C.; Park, C. Foraging trip duration of honeybee increases during a poor air quality episode and the increase persists thereafter. Ecol. Evol. 2021, 11, 1492–1500, doi:10.1002/ece3.7145.
- Gress, T.W.; Denvir, J.; Shapiro, J.I. Effect of Removing Outliers on Statistical Inference: Implications to Interpretation of Experimental Data in Medical Research. Marshall J. Med. 2018, 4, doi:10.18590/mjm.2018.vol4.iss2.9.
- Rodney, S.; Kramer, V.J. Probabilistic assessment of nectar requirements for nectar-foraging honey bees. Apidologie 2020, 51, 180–200, doi:10.1007/s13592-019-00693-w.
- Tosi, S.; Burgio, G.; Nieh, J.C. A common neonicotinoid pesticide, thiamethoxam, impairs honey bee flight ability. Sci. Rep. 2017, 7, doi:10.1038/s41598-017-01361-8.
Reviewer 2 Report
My compliments, This is an excellent study that has been performed well in all its aspects and with relevant parameters and with very interesting outcomes. Not relevant for this study is that the larval rearing method has been developed and tested to record larval/ pupal mortality. I did not see the overall success of the larval rearing. As said, not relevant for this study but could be interesting to show the success of this rearing method.
Author Response
1. My compliments, This is an excellent study that has been performed well in all its aspects and with relevant parameters and with very interesting outcomes. Not relevant for this study is that the larval rearing method has been developed and tested to record larval/ pupal mortality. I did not see the overall success of the larval rearing. As said, not relevant for this study but could be interesting to show the success of this rearing method.
Thank you very much for your feedback. We have the survival data and achieved a survival rate of approximately 80% to 90% from feeding until emergence with slight not significant differences between the replicates. However, we agree that this data is not relevant for the study, which is why we did not include it in the manuscript.